# "If you miss that first step in the chain of survival, there is no second step"–Emergency ambulance call-takers' experiences in managing out-of-hospital cardiac arrest calls

Nirukshi Perera[1,2]*, Tanya Birnie[1], Austin Whiteside[1,3], Stephen Ball[1,3], Judith Finn[1,3,4,5]

1 Prehospital, Resuscitation and Emergency Care Research Unit (PRECRU), School of Nursing, Curtin University, Bentley, Western Australia, Australia, 2 Institute for Communication in Health Care, School of Literatures, Language and Linguistics, Australian National University, Canberra, Australia, 3 St John Western Australia, Belmont, Western Australia, Australia, 4 Department of Epidemiology and Preventive Medicine, Monash University, Victoria, Australia, 5 Emergency Medicine, The University of Western Australia, Crawley, Western Australia, Australia

* niru.perera@curtin.edu.au

**Data Availability Statement:** All relevant data are within the paper and its Supporting Information files.

## Abstract

When a person has an out-of-hospital cardiac arrest (OHCA), calling the ambulance for help is the first link in the chain of survival. Ambulance call-takers guide the caller to perform life-saving interventions on the patient before the paramedics arrive at the scene, therefore, their actions, decisions and communication are integral to saving the patient's life. In 2021, we conducted open-ended interviews with 10 ambulance call-takers with the aim of understanding their experiences of managing these phone calls; and to explore their views on using a standardised call protocol and triage system for OHCA calls. We took a realist/essentialist methodological approach and applied an inductive, semantic and reflexive thematic analysis to the interview data to yield four main themes expressed by the call-takers: 1) time-critical nature of OHCA calls; 2) the call-taking process; 3) caller management; 4) protecting the self. The study found that call-takers demonstrated deep reflection on their roles in, not only helping the patient, but also the callers and bystanders to manage a potentially distressing event. Call-takers expressed their confidence in using a structured call-taking process and noted the importance of skills and traits such as active listening, probing, empathy and intuition, based on experience, in order to supplement the use of a standardised system in managing the emergency. This study highlights the often under-acknowledged yet critical role of the ambulance call-taker in being the first member of an emergency medical service that is contacted in the event of an OHCA.

**Funding:** This study was funded by a research grant from the Curtin School of Nursing. SJ-WA provided support in the form of in-kind participation for co-author AW and by allowing call-taker participants to undertake the interviews during their work shift. SJ-WA did not have any additional role in the study design, data collection and analysis, decision to publish, or preparation of the manuscript. The specific role of AW is articulated in the 'author contributions' section.

**Competing interests:** We have read the journal's policy and the authors of this manuscript have the following competing interests: AW is an employee of SJ-WA; SB and JF hold adjunct research positions with SJ-WA and JF receives research funding from SJ-WA. This does not alter our adherence to PLOS ONE policies on sharing data and materials.

## Introduction

Out-of-hospital cardiac arrest (OHCA) is a research focus in the field of emergency medicine and prehospital care due to its highly time-critical nature. When a patient has a cardiac arrest, outside of the hospital context, immediate intervention, usually by a lay bystander, is imperative because the chance of survival otherwise decreases by up to 10% per minute [1]. To this end, the "chain of survival" was conceptualised to outline the time-sensitive interventions that must be optimised in order to maximise the chance of survival [2]. The chain of survival includes four key components: 1) early recognition and the call for help; 2) early cardiopulmonary resuscitation (CPR); 3) early defibrillation; and 4) post-resuscitation care [2]. This article focusses on the first link in terms of members of the public accessing the emergency medical service (EMS) via the phone, and how that call is handled by emergency medical call-takers in the ambulance call centre.

In the last decade, research concerning the first links in the chain of survival took a turn to focus on detailed analyses of interactions between the caller and call-taker, based on audio recordings of the emergency phone call. Such research has helped improve our understanding of how callers answer the critical "is s/he breathing?" question [3], how they comprehend different prompts used in standardised call-handling scripts [4]; and how call-takers encourage caller compliance when they initiate CPR [5]. Research that focuses on the emergency call interaction has also highlighted the influence of critical issues in OHCA emergency calls such as language barriers [6, 7] and emotion [8, 9]. However, much of this research takes a quantitative approach to analysis. One aspect of the OHCA call-taking process that needs more attention is the workforce who have the closest and most extensive contact with the very emergency calls that the aforementioned research is based on. In other words, there is a need for call-takers to be asked for their insights into the management of OHCA calls, as they play a pivotal role in directing bystanders to help the patient in those crucial minutes before paramedics arrive.

The lack of attention paid to call-takers' experiences in OHCA calls may be reflective of what Willis et al. [10] call the "invisibility of call-taker work". As Coxon et al. [11] state, call-takers do not have direct physical contact with the public (unlike paramedics) so they are often overlooked and under-researched. The researchers found that while call-takers express pride in the central role they play in medical emergencies, they felt "faceless", undervalued by others, and burdened by the workload [11]. The bulk of the research that does exist on call-takers concerns the impact of emergency medical dispatch work on call-takers' stress and wellbeing [10, 12, 13]. Qualitative inquiry has identified issues such as posttraumatic stress disorder (PTSD) [14] and vicarious trauma (when call-takers are not at the scene of the incident) through cumulative exposure to emergency events that they are unable to control [15]. This increased awareness of the context call-takers are operating in is reflected in policy improvements. In fact, in 2018, the Parliament of Australia held an inquiry into the role of government in addressing the high rates of mental health conditions experienced by first responders, emergency service workers and volunteers, resulting in 14 recommendations to improve the mental health support for first responders (for all types of emergencies, not only medical) [16].

More recently, studies have appeared that use qualitative interviews to elicit how call-takers experience emergency calls, i.e. how they perceive the interaction with the caller [17, 18]. Torlén Wennlund et al. [18] identified that call-takers have to attentively manage a multifaceted interactive task in which they utilise creativity to gather critical information, and continuously process and assess complex information. Looking at OHCA calls specifically, Bång et al. [19] conducted interviews with call-takers to learn how they perceive the experience of providing dispatcher-assisted telephone CPR. The findings were similar with those of Torlén Wennlund

et al.'s [18] study in that they reflected the complexity of the call-taking task, as well as the need for flexibility in order to adjust to the exigencies of the situation. Call-takers emphasised the emotional support aspect of their work, with the requirement to mentally connect with the caller and maintain composure [19]. Our study builds on this work by describing the experiences of ambulance call-centre staff in handling all phases of out-of-hospital cardiac arrest "000" emergency calls in one particular ambulance call centre. Furthermore, the study was novel in its aim to understand call-taker views about using a standardised call protocol and triage system for OHCA calls.

## Materials and methods

### Study setting

St John Western Australia (SJ-WA) provides emergency road ambulance services in the state of Western Australia (WA), servicing a population of approximately 2.6 million people in both metropolitan and rural locations [20]. All emergency services in Australia are accessed by dialling "000", a national number which is handled by Telstra, a telecommunications company. Callers must state whether they want police, fire or ambulance services and the town and state they are located in. Telstra then transfers the call to the appropriate emergency centre. In the case of ambulance services in WA, calls are transferred to the SJ-WA State Operations Centre (SOC) located in Perth.

During the study there were approximately 90 call-takers employed at the SJ-WA call centre. Call-takers at SJ-WA are not clinically trained. New recruits undertake a six-week training programme followed by eight weeks of supervised call-taking before they become independent call-takers. As part of their induction, they become registered with the International Academies of Emergency Dispatch (IAED), which requires that call-takers undertake ongoing training for biannual recertification. The IAED is closely linked to the Medical Priority Dispatch System™ (MPDS, version 13.3 at the time of the study) [21] which is the structured call-taking system used by SJ-WA. MPDS and SJ-WA treat OHCA calls as highest-priority emergencies, with SJ-WA dispatching "lights and sirens" and a multi ambulance resource response. In 2020, there were 2,698 confirmed OHCA cases managed by the call-centre [22]. Fig 1 outlines the OHCA call process according to the MPDS protocol. Generally there are two clinicians (known as a clinical support paramedics) available for consultation on each shift in the SJ-WA call centre. Some call-takers were additionally qualified to perform other roles in the call centre such as managing the dispatch of ambulance resources however the call-taking role itself did not include this task.

### Methodological approach

We used a qualitative study design, based on interviews with ambulance emergency call-takers to understand their experiences of managing OHCA calls. We adopted the method of reflexive thematic analysis (TA) to analyse the interview transcripts as it facilitates the identification of recurring themes that arise in the data [23]. The reflexive aspect of TA involves the researchers identifying and interrogating assumptions made about the data, as well as being reflexive about the analytic process [24]. We were guided by a realist/essentialist approach to understanding the call-takers' experiences, meaning that we treated participants' experiences and narratives as real to them, and thus, based our understanding on the language they used to reflect their reality [25]. The TA was inductive—aiming to stay as close as possible to the meanings in the data, and semantic—focussing on ideas that were explicitly stated [25].

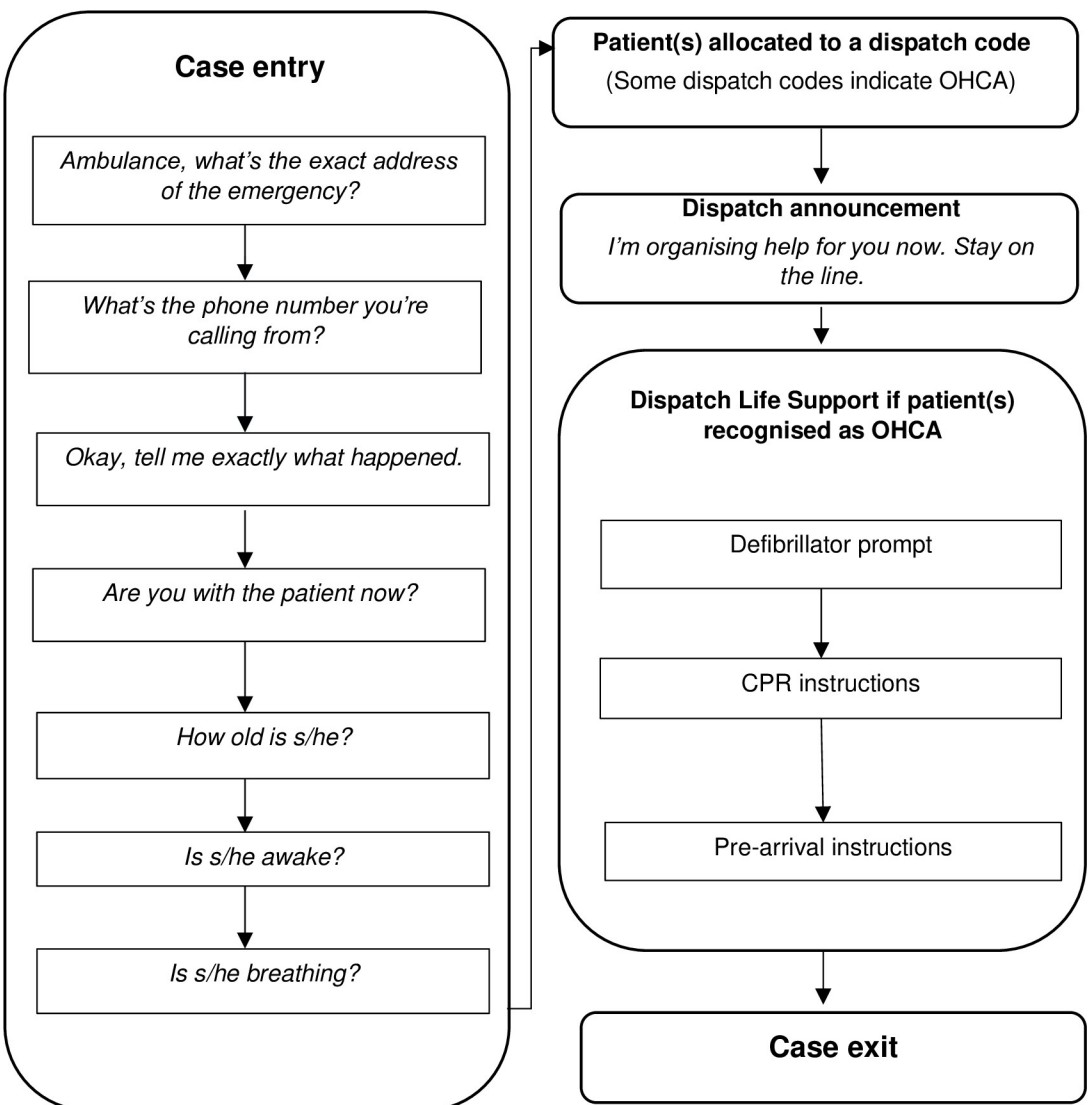

**Fig 1. The MPDS version 13 [21] call protocol for patients recognised as out-of-hospital cardiac arrest.**

## Participants and data collection

A series of one-on-one, face-to-face, in-depth interviews with the call-takers was conducted at the SJ-WA call centre between July and September 2021. The "in-depth" aspect of the interviews refers to an open-ended invitation to the participant to reveal as much as they can about the topic of interest [26]. The interview format was designed based on "unstructured" and "semi-structured" design principles. The unstructured component allowed for a few lead-in, open-ended questions to elicit free-flowing accounts from each participant. We also drew from semi-structured interview principles to allow for some prompts by the interviewer to assist deeper elicitation about issues that the participant raised [26]. However we aimed to treat the interviews as exploratory to see what topics the participants chose to discuss around the issue of managing OHCA calls and using the standardised protocol, rather than directing the interview towards pre-determined issues. (See S1 File for the interview format).

Purposive sampling involved calls for volunteer participants (see S2 File for flyers) which were distributed to all call-takers via email by AW, the Operations Manager of SJ-WA SOC. A lead-in time of three months for recruitment allowed for multiple notices, with incremental details about the study, to be sent to the target cohort so that call-takers had time to consider their involvement. Ethical considerations were paramount so that the call-takers, as employees of SJ-WA, did not feel pressured to participate in the study. Therefore, the call for participants emphasised that the research was an independent study to be led by NP, driven by her interest in call-taker communication, and that the research was not related to any evaluation of call-takers' employment performance at SJ-WA. Specific details about how confidentiality of participants would be maintained were also explained in detail, including the risks involved since participants would be attending the interviews during work time, thus their participation would not be entirely confidential. Participants provided written informed consent and chose a pseudonym for the interview that was only known by themselves and NP, the sole interviewer. The interviewer (NP) is a research fellow at the Prehospital, Resuscitation and Emergency Care Research Unit at Curtin University, has a doctorate in linguistics, has conducted several studies based on OHCA calls involving SJ-WA, and has experience in qualitative interview studies. The study was approved by the St John WA Research Governance Committee and the Curtin University Human Research Ethics Committee (HRE2021-0008 on 5 January 2021).

Anyone at SJ-WA SOC who volunteered to participate would be eligible for the study and we aimed to recruit a minimum of 10 participants to be consistent with recommended sample sizes for medium to large TA studies [25]. Interviews lasted between 30 minutes and 2 hours and were audio-recorded and then transcribed verbatim (assisted by transcription software [27]. After each interview, NP kept reflective field notes.

## Analysis

Once the transcripts were prepared, NP applied thematic analysis using NVivo software [28], following Braun and Clarke's [23] six phases of TA, which are 1) familiarising yourself with the data; 2) generating initial codes; 3) searching for themes; 4) reviewing themes; 5) defining and naming themes; 6) producing the report. NP recorded reflexive notes about the decision processes at each stage of the analysis in order to assist with refining the themes. At stage 5, TB audited the results for validation purposes, and after negotiation with NP, a consensus on the themes and assigned excerpts was achieved. At this point, NP sent a copy of the themes and relevant interview excerpts to each participant for their final approval of: 1) the analysis; 2) the publication of excerpts. A few participants requested for some potentially identifying words/ phrases to be omitted however those excerpts did not make the final selection for publication.

## Results

Ten volunteers (representing approximately one-ninth of SJ-WA's call-taker workforce) participated in the study. The participants' length of service ranged from a few to over 10 years.

The reflexive TA revealed four main themes regarding call-takers' experiences of managing OHCA calls: 1) time-critical nature of the call; 2) the call-taking process; 3) caller management; 4) protecting the self.

These four overarching themes were broken down into 14 sub-themes (Table 1). Each theme will be discussed below, with more attention being paid to the sub-themes that were most frequently discussed by participants (topics that were referred to by at least six of the 10 participants). Interview excerpts are provided to illustrate each theme. Note that information in [] is added by the authors for clarification.

**Table 1. Themes and sub-themes arising from call-taker interviews.**

| Themes | Sub-themes |
|---|---|
| Time critical nature of the call | Dealing with urgency |
| | Staffing |
| The call-taking process | The standardised protocol |
| | The computer interface |
| | Out-of-hospital cardiac arrest recognition |
| | Cardiopulmonary resuscitation |
| | Defibrillator retrieval |
| Caller management | Caller compliance |
| | Call-taker empathy |
| | Country calls |
| Protecting the self | Call follow-up |
| | Recognition |
| | Downtime and debrief |
| | Call-taker wellbeing |

## Time-critical nature of the call

The fact that OHCA is treated as the most time-critical medical emergency by EMS provided the context for the interviews and therefore generated much response. The two sub-themes under this theme were: 1) dealing with urgency; 2) staffing.

**Dealing with urgency.** Urgency characterises OHCA calls so call-takers provided their perspectives on how they handled this overriding influence in the phone interactions. The call-taker's experience of OHCA calls is aptly described in the quotes below from Deb;

> . . . you're knowing that if you don't get this right, by the time the crew gets there, they've got nothing to work with (Deb)

> . . . after a cardiac arrest, as soon as you see that pop up [that the patient is being taken to hospital], it's just exciting. I mean, you don't know what the results gonna be but they've got a chance . . . if I get like a lot of positive feelings after, it's usually more based on who the people on the scene were. So sometimes you get people who are super helpful, work together, and it's just like, wow, you feel so good about humanity. But if it was really difficult and people struggling to follow your instructions. Or maybe you're struggling to get information, it can be a bit draining. You kind of feel like you could have done more, could you have said more to get there, to encourage them. (Deb)

Deb's statements indicate the high level of investment call-takers had in OHCA calls. They were acutely aware of the importance of cooperative teamwork between the call-taker and the caller/bystanders to achieve survival outcomes.

Call-takers commented on the distinction between OHCA and other medical emergencies, where OHCA required immediate intervention from the bystander, and thus gave the call-taker a set process to implement with the caller;

> We're all just kind of treading water with most of those [other] jobs . . . We're stopping it getting worse, until someone can take them to hospital and fix them or fix them on scene. Whereas cardiac arrest is one of the few jobs we can step in very quickly and change an outcome. (Cody)

Call-takers exhibited keen awareness of the time-critical nature of OHCA calls and that rapid intervention was imperative;

This is not about preserving or being gentle and careful with him. This is a very vigorous thing we've got to do and we've got to do it quickly. Time now is of the essence. (Cody)

When call-takers became aware that the call was an OHCA emergency, they switched into a heightened mode;

I hate to say it, they all have a little bit of a thrill to them, because you get that adrenaline spike . . . you're amped up . . . your heart rate's up, you've got that adrenaline . . . and you're pounding through the call. (Alex)

Along with switching into a different mode, call-takers commented that they would change their voices to indicate urgency, although this could vary depending on the caller's situation;

I think it also depends how old the patient is. So if it's a newborn, you're a little bit more harsh on the caller, you need to say 'right now', 'we need to get this sorted'. Whereas if it's someone, they found dad and they think he might have been there a little while, you're a bit more like, 'okay, we're gonna get him on the floor' . . . a bit slower for them. (Kerry)

Call-takers spoke about the skills required to handle OHCA calls effectively. They had to be adept at multi-tasking while on the call;

You're clicking, you're moving, you're making sure the mapping, you're probably listening to somebody saying, you're doing 87 things at once. (Alison)

I think we put a lot on our call-takers to expect them to do all of these other things and arrange all these other resources. Or [the call-taker has to] ask other people to arrange these other resources whilst they're still trying to control a scene and have a system and follow a protocol. (Cody)

Cody referred to the fact that call-takers must, while initiating CPR, also be looking to see if there are any first responders or defibrillators close to the patient that could be brought in for assistance.

Scene management was also critical to the handling of the urgent OHCA call. Call-takers had to ensure the scene was accessible and safe for the paramedics. They also had to be alert to factors that would impact the emergency. Alison provided a scenario here;

Do we need to be addressing bleeding before even touching cardiac arrest? So sort of taking that time to determine the situation as a whole. Because we can get very 'okay they're not conscious, they're not breathing'. . . moving [quickly], and then you've not asked any more questions, because you've gone [on to CPR]. (Alison)

Critical thinking was a skill that many of the call-takers mentioned as necessary for the swift resolution of any obstacles that occurred during the call, especially when it came to barriers to commencing CPR. Prim pointed out that there were no visuals for call-takers, so listening became a sharpened skill;

They [paramedics] say when they go to scene, they assess, everything's visual. They'll look at coloured skin, they'll feel, they'll touch, they'll speak. For us, it's purely audio. So everything we hear becomes an absolute sixth sense almost. We actually go, 'that doesn't sound right'. And we pick up things that most people wouldn't even think to hear for or listen to. So [that's] our skill set and what we hear that people aren't saying. (Prim)

Call-takers discussed the balancing act they had to perform in managing callers' expectations and needs but also not giving them false hope;

We're not meant to say how far away the ambulance is. It's the dynamic environment, everything can change. Generally speaking, on those calls, because you've got multiple crews attending, I feel like we should be [able] to give a realistic expectation to say 'the ambulance is organised, help is on the way'. It's sometimes a bit unfair. If the ambulance is 30 minutes away, we need to be able to say, 'hey, they're 30 minutes away but we're going to do this' and set realistic expectations and realistic points of measurement. (Jane)

Call-takers outlined the difficulty of their position—to know when there were ambulance crews heading to the patient, but to be limited in the reassurance they could give the caller, as per the call centre policy. At the same time they knew this was the one piece of information that could have positive benefits for the initiation or continuation of bystander-CPR.

**Staffing.**   As part of their heightened awareness of the urgency of OHCA calls, call-takers expressed concern about whether the call centre had the capacity, that is, the quantity of staff to handle an increased volume of emergency medical calls. A shortage of staff on a shift impacted on call-takers' sense of control over their abilities to provide immediate attention to OHCA calls;

. . . understaffed in regards to our work volume has gone up quite significantly . . . they're [managers] just sort of playing catch up at the moment by recruiting a lot of new people. Just because the workload's gone through the roof. (Buddy)

For call-takers, watching the call queue grow, and knowing that they might not be answering an OHCA call in a timely manner, was a source of stress and frustration;

. . . we sit there watching the triple zeros across the bottom of screen. And then like five minutes later, cardiac arrest. You've already lost your window. You know, how long have they waited with triple zero? (Charlotte)

Charlotte raised the issue of surge capacity (not currently available at SJWA), so that non-call-taker staff at the ambulance headquarters could assist in peak periods;

There's no one to bring in to answer the calls. Other places have a surge capacity where people [other roles in the headquarters] . . . could be trained up. They should be trained up as call-takers. . . . bring in anyone and everyone to answer those triple zero calls. (Charlotte)

## The call-taking process

Various aspects of the call-taking system were raised by interview participants. In this section, five sub-themes are presented, two which are specific to the protocol used by the EMS: 1)

MPDS; 2) ProQA, and three concerning critical points in the OHCA call: 3) OHCA recognition; 4) CPR; 5) defibrillator retrieval.

**The standardised protocol.**   When it comes to managing OHCA calls, the overwhelming attitude about the dispatch system, MPDS, was positive. Call-takers pointed out that the standardised protocol equipped them with a set of steps to follow verbatim, thus providing structure to handle this critical emergency;

> I find cardiac arrest actually the easiest because that's when the protocol actually works. And you follow it, word for word. (Charlotte)

Cody pointed out that this structure meant that callers were assured a consistent service;

> So at least this gives a level playing field to everyone who calls up. Whoever they get who answers the phone, it's not luck that you've got someone that has got the background knowledge that knows what's going on. (Cody)

The phrasing in the script for OHCA calls also enabled swift progression to CPR and to instil urgency in the caller;

> The scripting within MPDS is very deliberate. It's very to the point. And the way that it is written does help convey that sense of urgency that you really need to get across to a caller. (Alex)

**The computer interface.**   When it came to processing OHCA calls, participants were generally positive about ProQA [29], the computer software package that accompanied MPDS. However they also acknowledged that not all call-takers found the system to be user-friendly. Several commented that the more recent recruits tended to favour ProQA whereas longer-serving call-takers found it harder to adapt from previous methods of manually entering case records.

Participants commented on the system enabling navigation between different parts of the OHCA call process;

> I find the ProQA system pretty awesome for me because . . . I like being more structured in a way. And everything's there as long as I'm clicking in the right [place] and following the right processes. It does make it a lot easier. (Buddy)

Deb pointed out that the system for OHCA allowed for a scenario where call-takers could move quickly to CPR instructions, and fill in other patient details (on their computer screens) once the caller had commenced compressions;

> It's a pretty good system. So it allows us to jump straight ahead. So we don't have to get every detail beforehand, that we can get to those compressions very early on, which I think is great. (Deb)

However some call-takers found that navigation through ProQA could be cumbersome;

> [what] I find very frustrating with MPDS is the multiple windows that you're jumping in and out of. Because we've got our call card on one. You try and update the crew as well, because the Fast Track doesn't give the crew an age. And they need to know what age the

patient is that they're going to. So you're jumping between MPDS and then your call card as well. (Jane)

Call-takers pointed out that ProQA provided a systematic record of their actions in the OHCA call which meant that they were protected in the event of an investigation;

> . . . it's also a system that protects us. If we follow it as our role and our job and our policy, well we're governed by that. So if there is any reason as to why . . . you know, things go to coroner's court and all that sort of stuff. 'What did she do?' Because they go from the first phone call to the ambulance to the ambulance attendants. So it's a whole research thing. 'Did we follow ProQA? Yes, done.' (Prim)

This, in fact, was a positive aspect of using a standardised protocol and system for call-takers because they felt safeguarded by it.

**Out-of-hospital cardiac arrest recognition.** The issue of timely recognition of OHCA during the call received much attention from the participants, raising issues about the callers mistaking ineffective breathing for effective breathing and the difficulty in using a breathing diagnostic tool in MPDS. In the following excerpts, call-takers spoke about the uncertainty in determining an OHCA if the caller was not clear about the symptoms.

Prim outlined a common scenario where call-takers could pick up clues of ineffective breathing even if the caller said the patient was breathing;

> . . . for example, if someone's called up and . . . someone's not breathing effectively, to the layman person, they'll be like, 'oh, you know, they're snoring or they're making a gasping [noise], they sound like a fish'. All these words to us are very important, but to them, they're still breathing. Because according to the caller, they're taking breaths in, but they don't realise that it's not sufficient. (Prim)

To alleviate some of these issues, many participants expressed taking a precautionary approach of treating uncertain statements of breathing as ineffective in order to trigger an OHCA response, even if the evidence was inconclusive;

> I would rather organise a dual response [two ambulances] or send it out as that response. And then start CPR and then go oh, 'actually, no, wait, hang on'. (Alison)

Prim highlighted the importance of call-taker experience in being able to detect the, sometimes non-verbal, signs of OHCA;

> 'They're dead' or 'they're not awake', 'they're not breathing'. They don't use words like that . . . 'they're asleep, but they're not responding'. . . so you have to use your experience, going, 'okay, so what are we going to do with this information? Are we going to go down the path of they're unconscious and assess, or [are we] going to find out more there and then?' . . . what I think is with cardiac arrest specifically, in this role, the more you do it, the more you pick up on what people say and what you hear. And it's more so what you hear that gives you the [conclusion], this is going to be a cardiac arrest, and then you start moving in that direction. (Prim)

Charlotte referred to the need for call-takers to overcome hesitancy in their assessment of a patient's breathing as this could waste time;

. . . it's that ambiguity and that hesitation that we have sometimes and maybe checking the breathing [again] when they've clearly given you one of the statements that you could just go into CPR. So I think that a lot of it is around training and education. (Charlotte)

Participants commented on the need for call-takers to do more probing in order to elicit the right information that could help to rule out or determine an OHCA;

. . . we prioritise based on the questions we ask . . . But like they say, we're only as good as the information provided. It's also up to us to make sure we get the right information given to us . . . so push for more if you're not happy. (Prim)

The MPDS dispatch system has a breathing diagnostic tool which call-takers can employ to assess if a patient is breathing effectively. This tool was frequently raised as a challenge to use with callers, as call-takers had to ask the caller to "say 'now' every time s/he takes a breath in starting immediately". Cody specified how the phrasing of the instructions could be changed to clearly instruct the caller to check the patients' inhalations;

Some of the scripting in the breathing assessment tool is difficult and people forever misunderstand what you want them to do. 'Okay I want you to tell me every time they take a breath starting now' [call-taker]. 'What've I got to do?' [caller] . . . So probably a preamble of 'okay, I'm going to see how their breathing is. What I want you to do is look at them closely. Watch the chest or the tummy rising and falling. Tell me when you've got that' [call-taker]. 'Yep, yep, I can see it now' [caller].' Okay, now tell me every time they take a breath' [call-taker]. Something like that. (Cody)

**Cardiopulmonary resuscitation.**   One of the challenging aspects of managing OHCA calls was when callers were resistant to initiating CPR. Call-takers were trained in how to encourage the caller to commence CPR and each had their individual approach to providing reassurance to the caller;

Reminding them that we're doing this together . . . saying, 'if you don't know how to do it, that's okay, I'm going to be here with you. I'm going to tell you'. Those kind of statements so that they don't think they're on their own. And that it's okay to try. Like I had one today where they were scared, they're gonna get blamed if something's wrong. (Deb)

For call-takers, it was sometimes difficult to gauge when they should push the caller and when to accept the caller's refusal. Call-takers showed understanding for refusal in certain circumstances;

If they refuse to do CPR, we can't force them, we just have to try . . . and that I thought was kind of nice, because she's obviously in her 90s. I think she had quite bad COPD [chronic obstructive pulmonary disease] and a few other health issues. And he's obviously thought, 'I just want her to go peacefully', as opposed to dragging her on the floor and doing CPR. (Kerry)

The participants commented on how initiating CPR was not only important for the patient but also for getting the caller to focus on an action rather than the panic they may be experiencing;

. . . because they're actually doing something. You know, 'I'm helping' or 'I'm saving this person's life' or whatever. It kind of shifts that focus away from 'oh my God, my loved one's dying' or 'this person is dying'. (Buddy)

In this excerpt, Kerry proposed that commencing CPR has a potential benefit to the caller of knowing they tried to save the patient rather than being a helpless onlooker;

Just sort of doing something, even though it obviously doesn't do that much. But at least you're trying something. And I also think sometimes we do CPR more, also for the person calling as well. Because if your loved one does die, wouldn't you like to think you tried to do something? . . . So they can be like, 'I did everything I could to save them' rather than just staring at the person. (Kerry)

In terms of the dispatch system, call-takers noted the difficulties in reciting a long set of CPR instructions to the caller. Call takers provided full "telephone-CPR" instructions, however if the caller was experienced in CPR or a medical professional then call-takers generally felt confident to let them handle CPR by themselves with support to ensure cadence and depth. The length of the instructions meant that callers sometimes missed information;

. . . the biggest stepping stone for me is just bang!. . . heaps of instructions straight out, like in a stressful situation, you know? . . . they're probably not listening. (Buddy)

Some call-takers found their own strategies to deal with the quantity of CPR instructions, such as breaking the instructions into smaller components;

So I read it all out and then I do it in a way that they can understand the second time. I read it and be like, 'now move the sheets from blah, blah, blah'. . . it's like a step-by-step thing, rather than 'get them on the ground'. (Kerry)

The procedure of moving the patient on to the floor for the commencement of compressions was identified as another challenging aspect of the CPR phase of the call because the MPDS script could not account for all the possible obstacles to achieving this. Again, call-takers had to exhibit critical thinking to rapidly resolve the issue;

I tried to work with them and brainstorm if there were any ideas they had, that we could use to get him onto the floor. So sometimes you do have to divert, you do have to kind of think on the fly. (Alex)

All participants found the phrasing of instructions in MPDS for positioning the patient to be helpful, especially the fact that these instructions could be broken down into steps;

. . . those instructions are really good. I've had a few people that were absolutely just refusing to try and move them and 'I can't move them. I can't move them'. But then when I've told them the step-by-step process, they've been able to do it. So I guess maybe it's just putting it into like a bite-sized chunks for them. (Kerry)

Another aspect of CPR that was raised was how call-takers instructed callers to maintain the quality of CPR once it had commenced. Due to space limitations, this is detailed in S3.

**Defibrillator retrieval.** The existence of public access defibrillators received considerable attention because call-takers felt that the system around using them needed improvement. Several call-takers mentioned that connecting with other emergency services, that is, the police and fire departments, could help in getting defibrillators to the scene of an OHCA in a timely manner;

> I suppose having a bigger resource pile for our second crews and things . . . using WA POL [police] and Firies [fire department] and things like that, because we don't look very far beyond the first responder schemes. Our first responders that are set up by the organisation are fantastic but there's 101 other defibs and people out there, first-aid-trained, that we never use . . . some sort of alert system for cardiac arrest for the police or firies. (Cody)

In the above quote Cody refers to SJ-WA's first responder scheme which uses an app to alert registered first-aid-trained citizens to nearby emergencies, however this app does not connect with other emergency services.

Call-takers expressed the need for in-room assistance, i.e., teamwork within the call centre, when it came to locating and organising a public access defibrillator;

> I would make more emphasis on getting defibs to scene. So at the minute, some people in the room are really good at, if you see a colleague on a resus, I will, for example, click on the resus [case record], see if there's any defibs nearby. If there's one nearby, got a phone number, I'll say 'I'm going to call for the defib'. (Kerry)

As with CPR, call-takers displayed concern for the callers and bystanders when it came to encouraging defibrillation, that is, not wanting to put a bystander in danger by asking them to race off to find a defibrillator; and questioning whether it was better to focus the bystander on doing compressions;

> . . . there's a defib 100 metres down at the [petrol station], someone needs to go and get that. They don't want to leave that person. There's two people, . . . How far do we send them? Or how much should we push them to do that? . . . you're putting these people in a stressful situation in another stressful situation . . . trying to get one of them to leave the home or leave the resus. (Suji)

## Caller management

The third theme from the interviews concerned the callers–how call-takers managed caller compliance and emotions, ranging from aggression and hysteria to detachment, and how they factored the caller's circumstances into their decision-making during OHCA calls.

Call-takers saw caring for both parties as part of their responsibility;

It's about that management of your caller, as well as the patient. (Alison)

Call-takers identified OHCA calls in country (rural) locations as situations where caller management was critical;

> Country WA is very different to other states. And our responses can be really, really long times. And you need to develop a relationship with that caller over the phone while we're doing CPR. It's really important. Because you need to keep them going. (Jane)

If an OHCA occurred in a remote or country location, then call-takers had to spend a long time instructing and motivating callers to continue with CPR until the crew arrived. In such cases, empathy and support for the caller were seen as paramount;

> . . . your country ones are the ones that you bond with because you're spending longer on the phone. (Deb)

The two main sub-themes for caller management:1) caller compliance; 2) call-taker empathy, are summarised below.

**Caller compliance.** The key to successful management of an OHCA call was ensuring that the caller answered all questions clearly and promptly and complied with call-takers' instructions, in order for CPR to be immediate and effective. This was identified as a challenging aspect of OHCA calls;

> . . . controlling of the call is harder than going through the MPDS system of the cardiac hands-on-chest and everything. Getting them to comply is the hard part. (Suji)

To ensure compliance, call-takers had to be able to handle a range of emotions and behaviours from callers;

> And it is about that call management. I think if you take the time to make sure your emotion is managed and understand that even if the person is aggressive, or yelling at you, 'okay, this is an extremely stressful situation for that person, how can I be of assistance to them?' Not matching or trying to override them. 'How can I help the caller to help the patient?' Because I think taking that time makes the rest of the call a lot easier. (Alison)

Calming some hysterical callers required patience from the call-takers who had to try an array of strategies and skills to take control;

> We call them screamers. They're on the call, and they're just that high. And no matter what you say, no matter what you do, no matter . . . my training, my experience, I'm going to give it to you, and they just don't listen or they don't stop because they're just that upset . . . You can't bring them back so . . . you ask [for] someone else. 'Will you give the phone to someone else?' and get them away, or 'you shut the door', all those sorts of things. (Prim)

> You just have to be very firm. Almost to the extent that you have to say, 'shut up' without using that word. Try and get their stress level down. For me, one of the best things to do is try and get their name because then you can just say, 'okay Emma, okay Emma, okay Emma'. . . just reiterate 'you have to do this, you're doing the best thing that you can' . . . while you're waiting for the crew to get there. (Suji)

Apart from the highly emotional callers, call-takers also had to be alert to those who appeared compliant;

> They're [OHCA callers] actually usually the most compliant callers. They're one extreme or the other, they're off the Richter and it's really hard to get control of them. Or they're just compliant. The only thing you've got to be careful of if they're really compliant, is that you can hear them doing something . . . So a lot of what we have to do is make sure that we can either hear them doing it or get them to acknowledge that they understand and are doing it. (Charlotte)

Charlotte raised the issue that call-takers do not only rely on what callers tell them but what they can hear happening.

**Call-taker empathy.** While call-takers spoke of the difficulties of dealing with emotional callers, they also exhibited much empathy for the caller and patients' situations, as part of their commitment to fulfilling the demands of their role. Therefore, blame for the challenging situation was not directed at the callers;

> ... you can't expect the caller in that moment of panic to be problem solving. (Deb)

Alex noted that his role was not just about achieving a survival outcome for the patient but actually supporting the caller through the call;

> If we can help the caller and make them feel that they've done everything that they could to help dad. To help mum. To help their brother, their son, their daughter. If we help them through that process, I think that's just as important as what the outcome for the patient is. (Alex)

Along these lines, participants cited compassion as integral to how they handled the caller, especially when the caller was tending to a loved one;

> So I want to display the compassion to that caller that I would expect if I was the caller or one of my family's the caller. (Jane)

As part of their empathy for callers, participants were aware of the significance of this phone call in the callers' lives;

> This is a normal day for us but for the people calling in this is a very abnormal day. This is probably a situation they've never ever been in, and hopefully never going to be in again. (Cody)

Especially in the case where it is likely the patient would not survive, call-takers stated that callers' wellbeing would then become their primary concern;

> ... at least you've got the caller to do CPR, so you've helped them with their healing process, because now they know they've tried everything, and you were there for them when they needed to do [something] in that moment. And so it's almost like a way to think about it. If the patient dies, there's still possibly some good that can come out of it. (Kerry)

## Protecting the self

The need to protect oneself in dealing with traumatic incidents and call exchanges was a key theme in the interviews. Call-takers referred to issues that arose, not within the OHCA call itself, but after the completion of the call. They commented that the end of an OHCA call could be abrupt, with the call-taker having to counsel and coach the caller through CPR and then the call ending as soon as the paramedics arrived, sometimes with a hang up of the phone and no departing exchange between the caller and the call-taker. This was often followed by the call-taker immediately taking the next call in the queue.

Most participants mentioned that, after taking the call, they elected to track the OHCA case on the dispatch system to see if the paramedics took the patient to hospital (indicating hope of

survival) or if the case was recorded as a patient death. However, the information available on the system was limited so call-takers were left with wanting to know more about the outcome but also not wanting to be too invested, that is, they had to be satisfied with knowing they did the best they could to help the patient and caller;

> I think, to know what happened, or if there is a success story, or if there isn't . . . because at the end of the day we're also putting effort into these people. In their lives. And we just don't know what happens to them at the end of it. (Prim)

Call-takers were also in the difficult position of knowing there was often little recognition of their contribution yet still hoping for some acknowledgment for their sense of job satisfaction and affirmation of their fundamental role in potentially saving a patient. They were aware that they were "just a voice on the phone" (Cody). The call-takers did not necessarily want to be regarded as heroes but as playing a critical part in a team that helped to save someone's life;

> So you're really trying to do your side of it and your job really well. Because if you miss that first step in the chain of survival, there is no second step. We're just forgotten about so much. We're not thanked. You know, 'oh the paramedics are the heroes'. They are, they absolutely are. But who started that CPR? Who gave them something to work with? (Deb)

The two main sub-themes for protecting the self: 1) downtime and debrief; 2) call-taker wellbeing, are summarised below.

**Downtime and debrief.**   The intensity during the OHCA call, where call-takers had to exercise complete concentration to work swiftly and definitively to get the caller to commence resuscitation, followed by an abrupt end to the call, could be a taxing experience, both mentally and emotionally. Call-takers spoke about the importance of self-care, in taking a few moments to calm down after the call had finished, as part of their coping mechanisms;

> I think sometimes you do need a moment, at the end of call. Just a breath . . . settle that, take a moment and move on. And sometimes we don't always have the luxury in our room to be able to do that. And sometimes, at the end of the day, I might just take a moment to go 'cool that happened'. We all have different triggers for what bothers us more. (Alison)

Alex spoke about the strange nature of OHCA calls when a call-taker drops in and out of a critical moment in the caller's life;

> . . . and we go into these really intense periods where we have to stay calm. We have to stay in control . . . we're doing everything we can to maintain the scene and keep them going and keep them focused. . . . that snapshot in a person's life. And then it's just done. Fine, hangs up. And there's another call waiting. It's a very strange, surreal way to look at life and death. (Alex)

Call-takers expressed the need to have downtime and debrief as part of organisational policy, as they had seen done for paramedics;

> Sometimes, some big cases, they'll have debriefs [for the ambulance crews] here at SOC. And we won't get released from the room to be a part of it because it's too busy for us. Well you pulled in a road crew off when it's busy. (Deb)

Call-takers advised that there was a policy in place where, if a call-taker took a distressing OHCA call, then this would trigger the organisation's support team to check-up on the employee via email or phone. However call-takers said implementation of the policy tended to be inconsistent and they did not always have the opportunity to talk to the team during their shift if it was busy.

**Call-taker wellbeing.** OHCA calls had the potential to impact call-takers' sense of wellbeing. They pointed out that cardiac arrest calls tend to "hit you a little bit harder" (Alison) than other calls;

> . . . just the way that a caller talks about a patient or there's something about it that just niggles at you a little bit. And that can be hard. (Alison).

In the following quote, Deb describes the potential effect of an OHCA call on the call-taker, due to experiencing the high-pressured first moments of a patient's cardiac arrest;

> . . . just because you're not there doesn't mean it's not connecting [with you]. . . obviously they're not all [like that] but you do have just a much higher exposure to trauma. And, you know, you might not be there at that hanging that turns into a resus. But you're hearing the mum's scream when she finds her child. You're hearing all that first stuff. (Deb)

Over time in the job, call-takers said that they developed their own coping mechanisms, however, some call-takers felt that;

> we're not checking in on people after they take so many horror calls. Not really looking after them. St John does this concept of self-reporting to wellbeing and support and they do some notifications [call-taker sent a generic email from support team]. But we know we're in an industry where we won't self-report. We're not traditionally the kind of people that will go and ask for help. (Charlotte)

Charlotte noted that, in emergency services, seeking help was not always part of the culture. Cody saw this as an area that needed improvement;

> I think the follow up of 'you've just taken a cardiac arrest, are you okay?' Because . . . they are still not nice calls and the impact . . . we don't know what's going on in their family at the time . . . I think . . . that touching base with 'okay, do you want more information about this?' . . . 'no worries I'll get clinical onto it, and we'll let you know how this worked and what the crew found'. (Cody)

Several call-takers also referred to the fact that taking calls could be triggering in terms of reminding them of past or potential traumas with their own loved ones. In relation to PTSD, Suji suggested that support was required, not only for the immediate call, but also for holistic wellbeing maintenance;

> . . . it should be mandatory for everybody to go and see a psych. Like maybe once a year, once every six months. And the reason I say that is because I've had a couple of friends from work that have left because they've ended up with PTSD or issues with the job. (Suji)

## Discussion

Call-takers exhibited a high level of awareness, reflection and insight about their role in the chain of survival and the varying factors that impacted effective management of an OHCA call. It was evident that participants took their responsibilities to both callers and OHCA patients seriously. One aspect that was repeatedly mentioned was the need for sharpened listening skills, to not only listen to the nuance of what callers stated or omitted ], but also to the background noise at the scene. Without a visual reference, call-takers had to rely on close listening (also known as applied or active listening) [19, 30] as well as intuition which was developed over years of service–these traits were critical to the OHCA recognition and CPR phases of the call. Language and voice were also highlighted as important. Even though a script was supplied via the MPDS protocol, call-takers had to supplement this with words of motivation and reassurance, often thinking "on the fly" to overcome obstacles. Speaking with authority and firmness during OHCA calls was necessary in order to maintain calmness and compliance in the caller and control of the life-saving intervention for the patient.

Call-takers expressed profound compassion for the caller's experience in an OHCA call across all four themes. The overriding attitude was one of empathy for the caller, being acutely aware of the distress they were experiencing in the case of tending to a loved one. Call-takers were alert to the extra care needed for some OHCA calls such as those coming from remote and rural locations, because of the strain the prolonged ambulance response time put on the bystander performing CPR. This trait of empathy in call-takers has been consistently mentioned in several studies [18, 30], with Bång et al. [19] conceptualising the role as being the "empathic authority that relieves the witness of their burden of responsibility". To this end, call-takers displayed flexibility in catering to the specific conditions surrounding individual OHCA cases and adapted their approach and behaviour accordingly. The sense of care for the caller as well as the patient infiltrated into many aspects of the OHCA call from weighing up how much to pursue for CPR, to ensuring caller safety, and how the experience of witnessing this arrest would affect the caller in the long-term. In this way, call-takers likened their role in the chain of survival to that of the caller in that both roles were often not recognised as critical to OHCA survival.

Undergirding these experiences of the OHCA call was the call-takers' adherence to the standardised MPDS dispatch protocol. An unexpected finding of this study was that call-takers found the system to work well for them when it came to managing OHCA calls. Some issues within MPDS were identified, reflecting the findings of research that interrogates the communicative strengths of the scripting [4, 31]. However the overall view of the dispatch system's performance was positive. The system provided call-takers with consistent phrasing and structured steps to relay to the caller, and afforded them protection to perform their roles confidently. At the same time the participants emphasised the importance of using their experience, intuition and critical thinking to supplement the standardised system, when it came to handling the particularities of each case.

The extent to which empathy was experienced in their day-to-day work had repercussions for the call-takers. How to protect themselves from becoming emotionally and mentally overburdened was expressed as a concern. This is one aspect of handling OHCA calls where the EMS could intervene to assist in call-takers' wellbeing maintenance. Expressed needs included providing those call-takers, who wished to know, with more information on the outcome of an OHCA patient, as well as implementing policies to give call-takers guaranteed downtime and debrief sessions with ambulance crews after traumatic cases. Also, instead of requiring them to self-present to the support team, call-takers were looking for support that was embedded into call centre practices such as immediate debriefing after a distressing call or regular sessions

with a psychologist. Recent studies into the use of mindfulness-based interventions to help reduce call-taker stress offer another possible avenue to assist in integrating wellbeing practices into organisational behaviour [32, 33]. Furthermore, research has suggested that more targeted training for call-takers, that directly addresses their expressed needs, whether that be about the use of the standardised protocol or caller management, will assist in helping call-takers to cope with the demands of their role [11, 17].

The issue of staff shortages was identified as a key stressor for the participants in terms of their workloads and their capacity to assist OHCA emergencies in a timely manner. This particular issue was revealed as a long-term challenge for the EMS to suitably recruit and roster adequate numbers of staff to meet demand. Factors such as the COVID-19 pandemic and overuse of the "000" number [34] also contributed to the pressure put on the call centre.

Raising the profile of call-takers' roles in the chain of survival was also important to their sense of job satisfaction. Call-takers' invisibility [10] created a feeling of being the unsung heroes of the EMS, lacking acknowledgement for the fact that, without their initial intervention, there would potentially be no chance of OHCA patient survival. Such sentiments were not expressed in a disgruntled manner rather they were articulated out of pride for their work, and out of a need for change to ensure a safe working environment for the call-taker. As part of recognition for their key role, call-takers sought support so that individuals did not burn out and leave the organisation, as tends to happen in caring, patient-centred roles [11].

Interviewing the call-takers for their experiences of OHCA calls enables a more holistic understanding of the call-taking process by elucidating why call-takers behave in certain ways during the call interaction. EMS and researchers need to consider not only what occurs during the call interaction but also the factors that impact on call-takers' performance and their ability to handle an OHCA call effectively. Therefore qualitative research concerning "end-user" perspectives of the emergency dispatch system can help to achieve this by accessing the expertise of the call-takers to improve the process based on their first-hand experiences with OHCA calls. Therefore we recommend further qualitative investigation into the environmental, organisational and psychological factors that impact on call-takers' ability to manage time-critical emergencies and how such factors can be better addressed by EMS worldwide.

## Limitations

This study was able to recruit 10 participants for the interviews. A factor that limited the number of volunteers may have been call-takers' perceptions that their employment would be compromised if colleagues could identify their interview comments, given that one of the co-authors was a manager at the call centre. Although information was presented to counter this idea, it may not have been adequate to convince some potential participants. Therefore, it is acknowledged that the perspectives provided in this study may be attributable to a subset of call-takers at the EMS, that is, those that felt secure to comment on their work and the organisation.

Due to space limitations, only a small number of interview excerpts were selected for publication to be representative of the views of the participants. This meant that the entirety of perspectives and nuanced views about a theme could not be presented in this article. However a more detailed and comprehensive report on the findings was provided to the EMS and to the volunteer participants.

## Conclusion

This interview study uncovered the experiences of call-takers in handling OHCA calls and characterised their roles in the first step in the chain of survival as: critical and complex,

requiring multi-tasking, critical thinking, calmness, authority, self-control, empathy, and a high level of responsibility. These traits were supported by a standardised dispatch system that served the call-takers well when it came to rapid OHCA recognition, life-saving intervention, and protection for their decision-making. However call-takers also highlighted the lack of recognition and the impact of OHCA calls on their wellbeing as key issues that EMS need to address in order to support the vital role call-takers play in improving outcomes for, not only OHCA patients, but the callers and bystanders who assist them.

## Supporting information

**S1 File. Interview format.**
(DOCX)

**S2 File. Recruitment materials.**
(PDF)

**S3 File. Additional interview excerpts for cardiopulmonary resuscitation.**
(DOCX)

## Acknowledgments

The authors are grateful to the 10 call-takers for their participation in this study and for the provision of honest and insightful interviews. We thank Ria Hardy and Julian Smith, and duty managers at St John WA State Operations Centre, for their assistance with this data collection.

## Author Contributions

**Conceptualization:** Nirukshi Perera, Austin Whiteside, Stephen Ball, Judith Finn.

**Data curation:** Nirukshi Perera, Tanya Birnie, Stephen Ball.

**Formal analysis:** Nirukshi Perera.

**Funding acquisition:** Nirukshi Perera, Judith Finn.

**Investigation:** Nirukshi Perera.

**Methodology:** Nirukshi Perera, Judith Finn.

**Project administration:** Nirukshi Perera, Tanya Birnie.

**Resources:** Austin Whiteside, Judith Finn.

**Supervision:** Judith Finn.

**Validation:** Tanya Birnie.

**Writing – original draft:** Nirukshi Perera.

**Writing – review & editing:** Nirukshi Perera, Tanya Birnie, Austin Whiteside, Stephen Ball, Judith Finn.

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
