## [Decision Letter · Decision Letter 0]

1 Sep 2022

PONE-D-22-18966“Because if you miss that first step in the chain of survival, there is no second step” – Emergency ambulance call-takers’ experiences in managing out-of-hospital cardiac arrest callsPLOS ONE

Dear Dr. Perera,

Thank you for submitting your manuscript to PLOS ONE. After careful consideration, we feel that it has merit but does not fully meet PLOS ONE’s publication criteria as it currently stands. Therefore, we invite you to submit a revised version of the manuscript that addresses the points raised during the review process.

We look forward to receiving your revised manuscript.

Kind regards,

Rishabh Charan Choudhary

Academic Editor

PLOS ONE

Journal Requirements:

"I have read the journal's policy and the authors of this manuscript have the following competing interests: AW is an employee of SJ-WA; SB and JF are research associates of SJ-WA."

We note that one or more of the authors are employed by a commercial company: SJ-WA.

Reviewers' comments:

Reviewer's Responses to Questions

**Comments to the Author**

1. Is the manuscript technically sound, and do the data support the conclusions?

Reviewer #1: Yes

Reviewer #2: Yes

2. Has the statistical analysis been performed appropriately and rigorously? 

Reviewer #1: N/A

Reviewer #2: N/A

3. Have the authors made all data underlying the findings in their manuscript fully available?

Reviewer #1: Yes

Reviewer #2: Yes

4. Is the manuscript presented in an intelligible fashion and written in standard English?

Reviewer #1: Yes

Reviewer #2: Yes

5. Review Comments to the Author

Reviewer #1: Long title- Remove Because, Abstract OHCA after definition and then use OHCA for the rest of the abstract. 190 remove anyway 195 remove years 201 remove a further 203 remove that is 276 remove on. Discussion remove all of the first paragraph. 746 remove on the 780. However...781, call-takers relied782 remove them with and they also and add and 784 remove as mentioned above 789 call takers. How to.... Confirm version of English for publication center vs centre and organization vs organisation

Reviewer #2: An interesting and potentially useful contribution to the limited literature the role of call takers in OHCA. Issues for clarification:

1. Further information on processes within the control centre would be helpful in providing a context for the qualitative data:

i. It appears that call takers also act as dispatchers of resources - this would not be a standard arrangement in large call centres. Please clarify.

ii. Do call takers provide full 'telephone-CPR' instructions or merely prompt a caller to start CPR if they know how?

iii. Is a 'CISM (critical incident stress management)' type system in operation for support or is this a more ad hoc arrangement?

iv. Are calls recorded and reviewed? By whom and with what purpose / outcomes?

v. It would be helpful to have an idea of the number of OHCAs dealt with by the centre annually.

2. The authors stress the work done to obtain ethics approval, organisational permissions and to reassure staff about confidentiality. It is of note, however, that one author is a member of management and the organisational ethos may have had an impact on those who volunteered for interview or on what they said. This may be a significant bias - information on the number of volunteers who came forward and how the 10 needed for the study were selected would be helpful in understanding this possible bias. It seems striking that all volunteers were very experienced. Comprehensive reporting on the study was provided to management - how was this explored with volunteers?

3. The interview content seems to focus almost exclusively on cases where OHCA was established. There is little reflection on the uncertainty or diagnostic challenge that can arise in the many cases where OHCA is considered, besides the well reported concerns about establishing agonal breathing. What about the many cases where the call taker thinks that death is well established - is CPR always advised or is there discretion? How do the staff feel about making that choice?

4. The themes identified and explored provide some interesting insights into the role.

5. A key finding in this study (and others) is the importance of the absence of feedback to staff. To what extent does its absence contribute to some of the stresses described? Would the authors recommend changes in this process?

6. Some reflection on the role of future research would be helpful in the discussion. In particular, it would be helpful to consider how this data might inform EMS systems / research elsewhere.

6. PLOS authors have the option to publish the peer review history of their article (what does this mean?). If published, this will include your full peer review and any attached files.

Reviewer #1: **Yes: **Jennifer Bradley

Reviewer #2: **Yes: **Gerard Bury

---

## [Author Response · Author response to Decision Letter 0]

23 Sep 2022

Response to Reviewers

We have considered your feedback and modified our article accordingly. Thank you for helping us to refine and improve our arguments. Please see our detailed responses below which we hope you will find satisfactory.

Please note that the line numbers we refer to in our responses here correspond to those in the tracked changes version of the revised manuscript.

Reviewer #1: 

1. Long title- Remove Because, 

Authors’ response: done, we have changed the title to: “If you miss that first step in the chain of survival, there is no second step” – Emergency ambulance call-takers’ experiences in managing out-of-hospital cardiac arrest calls

2. Abstract OHCA after definition and then use OHCA for the rest of the abstract. 

Authors’ response: done 

3. 190 remove anyway 

Authors’ response: done

4. 195 remove years 

Authors’ response: done

5. 201 remove a further 

Authors’ response: done

6. 203 remove that is 

Authors’ response: done

7. 276 remove on. 

Authors’ response: done

8. Discussion remove all of the first paragraph. 

Authors’ response: done

9. 746 remove on the 

Authors’ response: done

10. 780. However...

Authors’ response: done

11. 781, call-takers relied

12. 782 remove them with and they also and add and

Authors’ response: As the reviewer pointed out, this sentence structure is problematic, so we have reworded it to (Lines 792-5):

The system provided call-takers with consistent phrasing and structured steps to relay to the caller and afforded them protection to perform their roles confidently.

13. 784 remove as mentioned above 

Authors’ response: done

14. 789 call takers. How to.... 

Authors’ response: done

15. Confirm version of English for publication center vs centre and organization vs organisation

Authors’ response: we have used UK English in this article and have searched the document for uses of US English spelling (such as “organization” and “center”) but have not found any instances.

Reviewer #2: 

An interesting and potentially useful contribution to the limited literature the role of call takers in OHCA. Issues for clarification:

1. Further information on processes within the control centre would be helpful in providing a context for the qualitative data:

i. It appears that call takers also act as dispatchers of resources - this would not be a standard arrangement in large call centres. Please clarify.

Authors’ response: We have added a sentence to clarify that dispatching was not part of the call-taking responsibilities under Materials and Methods/Study Setting/Lines 122-24:

Some call-takers were additionally qualified to perform other roles in the call centre such as managing the dispatch of ambulance resources however the call-taking role itself did not include this task.

ii. Do call takers provide full 'telephone-CPR' instructions or merely prompt a caller to start CPR if they know how?

Authors’ response: We have added a sentence to clarify that call-takers did not have to provide all instructions if the callers were experienced in CPR under Results/The call-taking process/Cardiopulmonary resuscitation/Lines 488-90:

Call takers provided full “telephone-CPR” instructions, however if the caller was experienced in CPR or a medical professional then call-takers generally felt confident to let them handle CPR by themselves with support to ensure cadence and depth.

iii. Is a 'CISM (critical incident stress management)' type system in operation for support or is this a more ad hoc arrangement?

Authors’ response: We had made a statement about this in the Results section but the Reviewer’s question has prompted us to reword it to be more explicit. This is located under Results/Protecting the Self/Downtime and Debrief/Lines 703-7.

Call-takers advised that there was a policy in place where, if a call-taker took a distressing OHCA call, then this would trigger the organisation’s support team to check-up on the employee via email or phone. However call-takers said implementation of the policy tended to be inconsistent and they did not always have the opportunity to talk to the team during their shift if it was busy.

iv. Are calls recorded and reviewed? By whom and with what purpose / outcomes?

Authors’ response: The EMS recorded and reviewed the calls and a quality management process was undertaken with staff. While the review process did attract comments from the participants, and we agree that it is a significant issue for call centre staff, the comments were about reviews of calls in general, not OHCA calls specifically. Therefore they were not included in the results as they did not correspond with the aim of the research.

v. It would be helpful to have an idea of the number of OHCAs dealt with by the centre annually.

Authors’ response: We agree that this is a good idea and have added a sentence under Materials and Methods/Study Setting/Lines 119-120:

In 2020, there were 2,698 confirmed OHCA cases managed by the call-centre (22).

2. The authors stress the work done to obtain ethics approval, organisational permissions and to reassure staff about confidentiality. It is of note, however, that one author is a member of management and the organisational ethos may have had an impact on those who volunteered for interview or on what they said. This may be a significant bias - information on the number of volunteers who came forward and how the 10 needed for the study were selected would be helpful in understanding this possible bias. It seems striking that all volunteers were very experienced. Comprehensive reporting on the study was provided to management - how was this explored with volunteers?

Authors’ response: We agree about the need to be very clear about the ethical management of the study. We aimed to outline the separation of roles between AW (employee of SJ-WA) and NP (the interviewer and analyst) in Materials and Methods/Participants and data collection (Lines 158-166):

Purposive sampling involved calls for volunteer participants (see supplementary materials for flyers) which were distributed to all call-takers via email by AW, the Operations Manager of SJ-WA SOC. A lead-in time of three months for recruitment allowed for multiple notices, with incremental details about the study, to be sent to the target cohort so that call-takers had time to consider their involvement. Ethical considerations were paramount so that the call-takers, as employees of SJ-WA, did not feel pressured to participate in the study. Therefore, the call for participants emphasised that the research was an independent study to be led by NP, driven by her interest in call-taker communication, and that the research was not related to any evaluation of call-takers’ employment performance at SJ-WA.

We have also included the recruitment materials in the supplementary materials as evidence of these ethical considerations.

We have added the point that anyone who volunteered to participate was interviewed (that is, there was no selection process or eligibility criteria for the interview. 10 people volunteered and all were interviewed). See Materials and Methods/Participants and data collection/Line 178:

Anyone at SJ-WA SOC who volunteered to participate would be eligible for the study and we aimed to recruit a minimum of 10 participants to be consistent with recommended sample sizes for medium to large TA studies (25).

We agree with the reviewer that it was striking that some volunteers had many years of experience. However a third had about 2 years’ experience, and a few had about 5 years’ experience. We stated years of service in broad terms in the Results/Lines 199-200 section to protect identity:

The participants’ length of service ranged from a few to over 10 years. 

We made an oversight in not stating that comprehensive reporting was provided to both the EMS management and the volunteer participants. This has been rectified in the Discussion/Lines 856:

However a more detailed and comprehensive report on the findings was provided to the EMS and to the volunteer participants.

In regard to the role of the co-author who is part of SJ-WA management (AW), the editors of PLOS ONE have asked us to provide more detail in our competing interests statement. We state: 

We have read the journal's policy and the authors of this manuscript have the following competing interests: AW is an employee of SJ-WA; SB and JF hold adjunct research positions withSJ-WA and JF receives research funding from SJ-WA. This does not alter our adherence to PLOS ONE policies on sharing data and materials.

In the funding statement, we state:

This study was funded by a research grant from the Curtin School of Nursing. SJ-WA provided support in the form of in-kind participation for co-author AW and by allowing call-taker participants to undertake the interviews during their work shift. SJ-WA did not have any additional role in the study design, data collection and analysis, decision to publish, or preparation of the manuscript. The specific role of AW is articulated in the ‘author contributions’ section.

3. The interview content seems to focus almost exclusively on cases where OHCA was established. There is little reflection on the uncertainty or diagnostic challenge that can arise in the many cases where OHCA is considered, besides the well reported concerns about establishing agonal breathing. What about the many cases where the call taker thinks that death is well established - is CPR always advised or is there discretion? How do the staff feel about making that choice?

Authors’ response: In the results/the call-taking process/Out-of-hospital cardiac arrest recognition, we cover the most frequent issues raised by the interviewees regarding recognition. In this section, there are comments from call-takers about the difficulty of detecting ineffective breathing if the caller statements are not clear. Therefore call-takers discussed the need for further probing, close listening for clues and keywords, and to not hesitate in triggering an OHCA response if they are uncertain about breathing status. There were also comments about the importance of experience in being able to tell when something was not right and the clumsy wording in the MPDS breathing diagnostic tool. 

With regard to obvious or established death, some call-takers did comment positively about a recent change in the EMS’ policy. A change to the definition of obvious death meant that call-takers were not mandated to commence CPR in cases where the patient had been dead for some time. These findings were reported back to the EMS and participants. 

We did not home in on “obvious death” in the article because, as stated in the Results section, we pay more attention to “the sub-themes that were most frequently discussed by participants (topics that were referred to by at least six of the 10 participants). Due to space limitations we were unable to cover all points made by the call-takers which we noted in the Discussion/Limitations (Lines 852-4): 

Due to space limitations, only a small number of interview excerpts were selected for publication to be representative of the views of the participants. This meant that the entirety of perspectives and nuanced views about a theme could not be presented in this article.

4. The themes identified and explored provide some interesting insights into the role.

Authors’ response: Thank you for your comment.

5. A key finding in this study (and others) is the importance of the absence of feedback to staff. To what extent does its absence contribute to some of the stresses described? Would the authors recommend changes in this process?

Authors’ response: When we discuss the absence of feedback in this article we refer to call-takers’ comments about lack of opportunities for follow-up on the survival outcomes for OHCA cases and lack of opportunities for the staff to have downtime and debrief when needed. These are outlined in Results/Protecting the Self. We connect this to call-taker stress in the Discussion section and recommend that the EMS change its policies (Lines 799-808):

The extent to which empathy was experienced in their day-to-day work had repercussions for the call-takers. How to protect themselves from becoming emotionally and mentally overburdened was expressed as a concern. This is one aspect of handling OHCA calls where the EMS could intervene to assist in call-takers’ wellbeing maintenance. Expressed needs included providing those call-takers, who wished to know, with more information on the outcome of an OHCA patient, as well as implementing policies to give call-takers guaranteed downtime and debrief sessions with ambulance crews after traumatic cases. Also, instead of requiring them to self-present to the support team, call-takers were looking for support that was embedded into call centre practices such as immediate debriefing after a distressing call or regular sessions with a psychologist.

6. Some reflection on the role of future research would be helpful in the discussion. In particular, it would be helpful to consider how this data might inform EMS systems / research elsewhere.

Thank you to the reviewer for this suggestion. We have added a recommendation for future research in the discussion about the value of qualitative research on/for/with call-takers (Lines 838-840):

Therefore we recommend further qualitative investigation into the environmental, organisational and psychological factors that impact on call-takers’ ability to manage time-critical emergencies and how such factors can be better addressed by EMS worldwide.

---

## [Decision Letter · Decision Letter 1]

8 Dec 2022

“If you miss that first step in the chain of survival, there is no second step” – Emergency ambulance call-takers’ experiences in managing out-of-hospital cardiac arrest calls

PONE-D-22-18966R1

Dear Dr. Perera,

We’re pleased to inform you that your manuscript has been judged scientifically suitable for publication and will be formally accepted for publication once it meets all outstanding technical requirements.

Kind regards,

Rishabh Charan Choudhary

Academic Editor

PLOS ONE

Additional Editor Comments (optional):

Reviewers' comments:

Reviewer's Responses to Questions

**Comments to the Author**

1. If the authors have adequately addressed your comments raised in a previous round of review and you feel that this manuscript is now acceptable for publication, you may indicate that here to bypass the “Comments to the Author” section, enter your conflict of interest statement in the “Confidential to Editor” section, and submit your "Accept" recommendation.

Reviewer #2: All comments have been addressed

2. Is the manuscript technically sound, and do the data support the conclusions?

Reviewer #2: Yes

3. Has the statistical analysis been performed appropriately and rigorously? 

Reviewer #2: N/A

4. Have the authors made all data underlying the findings in their manuscript fully available?

Reviewer #2: Yes

5. Is the manuscript presented in an intelligible fashion and written in standard English?

Reviewer #2: Yes

6. Review Comments to the Author

Reviewer #2: Thank you for responses to comments, which address most of the issues raised. Thank you for responses to comments, which address most of the issues raised. Thank you for responses to comments, which address most of the issues raised.

7. PLOS authors have the option to publish the peer review history of their article (what does this mean?). If published, this will include your full peer review and any attached files.

Reviewer #2: **Yes: **Gerard Bury

---

## [Editor Report · Acceptance letter]

14 Dec 2022

PONE-D-22-18966R1 

“If you miss that first step in the chain of survival, there is no second step” – Emergency ambulance call-takers’ experiences in managing out-of-hospital cardiac arrest calls 

Dear Dr. Perera:

I'm pleased to inform you that your manuscript has been deemed suitable for publication in PLOS ONE. Congratulations! Your manuscript is now with our production department. 

Kind regards, 

on behalf of

Professor Margaret Williams 

Academic Editor

PLOS ONE